# The CD44high Subpopulation of Multifraction Irradiation-Surviving NSCLC Cells Exhibits Partial EMT-Program Activation and DNA Damage Response Depending on Their p53 Status

**DOI:** 10.3390/ijms22052369

**Published:** 2021-02-27

**Authors:** Margarita Pustovalova, Lina Alhaddad, Taisia Blokhina, Nadezhda Smetanina, Anna Chigasova, Roman Chuprov-Netochin, Petr Eremin, Ilmira Gilmutdinova, Andreyan N. Osipov, Sergey Leonov

**Affiliations:** 1School of Biological and Medical Physics, Moscow Institute of Physics and Technology, 141700 Dolgoprudny, Moscow Region, Russia; lina-alhaddad@hotmail.com (L.A.); tai2509@yandex.ru (T.B.); smetaninanm@gmail.com (N.S.); annagrekhova1@gmail.com (A.C.); netochin@gmail.com (R.C.-N.); 2State Research Center-Burnasyan Federal Medical Biophysical Center of Federal Medical Biological Agency (SRC-FMBC), 123098 Moscow, Russia; 3Semenov Institute of Chemical Physics, Russian Academy of Sciences, 119991 Moscow, Russia; 4Emanuel Institute for Biochemical Physics, Russian Academy of Sciences, 119334 Moscow, Russia; 5FSBI “National Medical Research Center for Rehabilitation and Balneology”, Ministry of Health of Russia, 121099 Moscow, Russia; ereminps@gmail.com (P.E.); gilm.ilmira@mail.ru (I.G.); 6Institute of Cell Biophysics, Russian Academy of Sciences, 142290 Pushchino, Moscow Region, Russia

**Keywords:** non-small cell lung cancer, cancer stem cells, radioresistance, Rad51, p53, epithelial-to-mesenchymal transition

## Abstract

Ionizing radiation (IR) is used for patients diagnosed with unresectable non-small cell lung cancer (NSCLC). However, radiotherapy remains largely palliative due to the survival of specific cell subpopulations. In the present study, the sublines of NSCLC cells, A549IR (p53wt) and H1299IR (p53null) survived multifraction X-ray radiation exposure (MFR) at a total dose of 60 Gy were investigated three weeks after the MFR course. We compared radiosensitivity (colony formation), expression of epithelial-mesenchymal transition (EMT) markers, migration activity, autophagy, and HR-dependent DNA double-strand break (DSB) repair in the bulk and entire CD44high/CD166high CSC-like populations of both parental and MFR survived NSCLC cells. We demonstrated that the p53 status affected: the pattern of expression of N-cadherin, E-cadherin, Vimentin, witnessing the appearance of EMT-like phenotype of MFR-surviving sublines; 1D confined migratory behavior (wound healing); the capability of an irradiated cell to continue to divide and form a colony of NSCLC cells before and after MFR; influencing the CD44/CD166 expression level in MFR-surviving NSCLC cells after additional single irradiation. Our data further emphasize the impact of p53 status on the decay of γH2AX foci and the associated efficacy of the DSB repair in NSCLC cells survived after MFR. We revealed that Rad51 protein might play a principal role in MFR-surviving of p53 null NSCLC cells promoting DNA DSB repair by homologous recombination (HR) pathway. The proportion of Rad51 + cells elevated in CD44high/CD166high population in MFR-surviving p53wt and p53null sublines and their parental cells. The p53wt ensures DNA-PK-mediated DSB repair for both parental and MFR-surviving cells irrespectively of a subsequent additional single irradiation. Whereas in the absence of p53, a dose-dependent increase of DNA-PK-mediated non-homologous end joining (NHEJ) occurred as an early post-irradiation response is more intensive in the CSC-like population MFR-surviving H1299IR, compared to their parental H1299 cells. Our study strictly observed a significantly higher content of LC3 + cells in the CD44high/CD166high populations of p53wt MFR-surviving cells, which enriched the CSC-like cells in contrast to their p53null counterparts. The additional 2 Gy and 5 Gy X-ray exposure leads to the dose-dependent increase in the proportion of LC3 + cells in CD44high/CD166high population of both parental p53wt and p53null, but not MFR-surviving NSCLC sublines. Our data indicated that autophagy is not necessarily associated with CSC-like cells’ radiosensitivity, emphasizing that careful assessment of other milestone processes (such as senescence and autophagy-p53-Zeb1 axis) of primary radiation responses may provide new potential targets modulated for therapeutic benefit through radiosensitizing cancer cells while rescuing normal tissue. Our findings also shed light on the intricate crosstalk between autophagy and the p53-related EMT, by which MFR-surviving cells might obtain an invasive phenotype and metastatic potential.

## 1. Introduction

Lung cancer is the leading cause of cancer mortality worldwide. Non-small cell lung cancer (NSCLC) represents 85% of lung cancers [1], about 40% are unresectable [2], and the 5-year overall survival (OS) rate is about 15% [1].

Stereotactic body radiation therapy (SBRT) has emerged as an effective and well-tolerated treatment for early-stage NSCLC [3]. SBRT relies on a small irradiation field to precisely delivered high doses of radiation, typically above 10 Gy per fraction, to local tumors. However, during the therapy, tumors can undergo complex Darwinian evolution leading to a selective outgrowth of clones with phenotypic advantage within a given microenvironment or therapeutic context and thus acquire radioresistance [4].

The main factor related to radioresistance is the presence of cancer stem cells (CSCs) [5], also called tumor-initiating cells (TICs) [6], inside tumors. They reflect the cellular heterogeneity within solid tumors [7] and have rapid proliferation, migration, and role in the recurrence of cancer [6] and express specific markers and stem cell genes [8,9]. In general, CSC marker-positive cells (CD133+, CD44+, CD166+, aldehyde dehydrogenase (ALDH+), and epithelial cell adhesion molecule (EpCAM+)) exhibit a 100-fold increased capacity to initiate cancer [6].

One of the potential sources of stem cells is polyploid cells characterized by multinucleation and cell cycle arrest [10,11]. Polyploidy was shown as an adaptation mechanism that allows cancers to tolerate harsh treatment conditions and plays significant role in the drug-resistance of many cancers and cancer cell lines, including NSCLC [12,13]. The polyploid giant cancer cells (PGCCs) could reversibly enter the therapy-induced senescence, allowing them to survive therapy and later re-enter the cell cycle to form daughter cells of typical size and ploidy [14,15]. Therefore, the progeny of metabolically active PGCCs may contribute to cancer recurrence following anticancer treatment [12]. Moreover, PGCCs have been demonstrated to possess stem cell-like properties, as they express CSCs markers, form spheroids in vitro, and generate tumors in mice [16,17,18].

The plasticity of CSCs, a process highly dependent on the epithelial-mesenchymal transition (EMT) and associated with cell dedifferentiation, is responsible for metastases, relapses, radiation therapy (RT) failure, and a poor prognosis in cancer patients [19]. EMT is characterized by the loss of epithelial morphology markers (such as E-cadherin) and acquiring the mesenchymal markers (including N-cadherin and Vimentin). Thus, cancer cells undergoing EMT acquire invasive and metastatic properties. Notably, the radiation survived sphere cells express significantly higher levels of CSC markers (CD24 and CD44), nuclear β-catenin, and EMT markers (Snail1, Vimentin, and N-cadherin) compared to non-irradiated lung tumorsphere cells [7]. The CSCs demonstrated upregulated DNA damage responses (DDRs), including enhanced checkpoint signaling and recruitment of DNA repair proteins [20,21]. DNA double-strand breaks (DSBs) are the most critical DNA damage. Mis-repaired or unrepaired DSBs lead to chromosome aberrations or cell death, respectively [22,23,24]. In mammalian cells, DNA DSBs repair is carried out by two canonical mechanisms: fast but incorrect, non-homologous end joining (NHEJ) and slow, but relatively correct, homologous recombination (HR) and several alternative pathways [25]. The mechanistic steps of HR-dependent repair involve DNA end resection, strand invasion, homology search, and pairing, followed by new DNA synthesis [26]. NHEJ contributes to repair of about 80% of DNA DSBs [27]. Unlike normal cells, for which DNA repair’s accuracy is crucial, tumor cells require DSBs repair efficiency, but not its accuracy [28]. However, there is evidence that the radioresistance of aggressive tumors and CSCs is mainly due to the increased activity of HR. Overexpression of Rad51, an essential protein involved in DNA DSB repair by HR pathway, is associated with a more aggressive cancer phenotype and treatment resistance in various tumors, including ovarian, cervical, prostate, pancreatic, colorectal, and malignant gliomas [29,30,31,32]. Rad51 protein is the main element involved in the HR process [33]. Rad51 catalyzes the homology search and the strand exchange with a homologous sequence and ensures the accurate repair of the DSBs. Moreover, it interacts directly with protein suppressors of breast cancer (BRCA1, BRCA2) [34] and p53 [35], which also indicates the importance of Rad51 in apoptosis. In response to DNA damage, the nuclear foci usually contain Rad51, and recombination proteins are microscopically detected as a surrogate for ongoing intracellular processes of DNA repair. 

Autophagy has been initially attributed to both tumor-suppressive and tumor-promoting functions [36]. Recent studies imply a role for p53 in regulating autophagy, a catabolic pathway by which eukaryotic cells degrade and recycle macromolecules and organelles, particularly under conditions of nutrient deprivation [37]. However, the autophagy’s role and extent in radiosensitivity related to p53 status of CSC-like NSCLC cells after multifraction radiation exposure (MFR) is not comprehensive and still in the focus of current research.

In our previous study [38], exponentially growing A549 (p53wt) and H1299 (p53null) cell lines were multiply irradiated by increasing doses of X-ray (ten fractions of 2 Gy, four fractions of 5 Gy, and two fractions of 10 Gy). The original parental cells were cultured without irradiation under the same conditions. The cells that survived after irradiation with a total dose of 60 Gy and gave rise to clonogenic growth were named A549IR and H1299IR sublines. All experiments were carried out on cells cultured for three weeks after irradiation. We demonstrated that in the presence of wild-type p53, the kinetics of each DSB marker (γH2AX and pATM foci) in A549IR cells were almost similar to that of parental cells. In contrast, in the absence of p53, the total number and kinetics of these DSB markers in irradiation-surviving H1299IR cells were significantly lower than that of parental cells during the early phase of DNA DSB response (within 8 h after additional 2 Gy X-ray test-exposure). Moreover, our data indicated more efficient IR dose-dependent 53BP1 recruitment to damaged chromatin and the 53BP1-mediated NHEJ in H1299IR cells compared to A549IR cells 24 h after single 2–6 Gy X-ray exposure. Nevertheless, it was still unclear whether the status of p53 of irradiation-surviving cells affects the HR-dependent DNA DSBs repair in response to ionizing radiation (IR) exposure.

Other studies also proved a difference in NSCLC cells’ sensitivity to radio- and chemotherapy depending on their p53 status [21,39,40,41]. However, the role of p53 in ensuring the resistance of NSCLC to MFR has not yet been fully elucidated.

In the present study, we investigated the effect of MFR on colony formation, expression of EMT markers, autophagy, migration activity, and HR-dependent DNA DSBs repair in CD44high/CD166high CSC-like populations of the parental (p53wt A549 and p53null H1299) as well as MFR-surviving (A549IR and H1299IR) NSCLC cells.

## 2. Results

### 2.1. Radiosensitivity of A549IR and H1299IR Cells

To determine the radiosensitivity of A549IR and H1299IR cells, we performed a colony formation assay. The results in Figure 1a demonstrate that A549IR cells had increased survival fractions compared with parental A549 cells. On the contrary, H1299IR cells were more radiosensitive compared with parental H1299 cells. Considering the decreased plating efficiency of H1299IR cells observed in our previous study [38] and their reduced ability to adhere to plastic, we performed an anchorage-independent soft agar assay. In fact, the soft agar assay results demonstrated reduced radiosensitivity of H1299IR cells. (Figure 1d). Collectively, these results indicated controversies in investigating of cancer cells radiosensitivity by routine colony formation assay. Irradiated cancer cells can lose adhesion ability but save ability to divide.

### 2.2. DNA Repair Capacity of Parental Cells and Their MFR-Surviving Sublines Depending on Their p53 Status

To evaluate HR’s contribution depending on the status of p53, we conducted a comparative analysis of the kinetics of γH2AX and Rad51 foci in parental and MFR-surviving sublines of NSCLC after additional single irradiation at a dose of 2 Gy. The cells were fixed 1–24 h after irradiation. Non-irradiated parental and radioresistant cells were used as controls. Representative immunofluorescent images of the irradiated cells showing Rad51, γH2AX foci and their colocolization are presented in Figure 2a.

We observed that in A549 and A549IR cells, there was a decrease in the foci number of γH2AX by 70–80% of the initial maximum 8 h after irradiation, and after 24 h almost reached the control level (Figure 2b). Moreover, at 8 h after irradiation, the γH2AX foci number in H1299 and H1299IR cells decreased by 65% and 50% of the initial maximum, respectively (Figure 2d). Simultaneously, the number of Rad51 foci changed insignificantly, and their maximum number was observed 4–6 h after irradiation in all populations of the NSCLCs (Figure 2c,e).

We assessed HR’s contribution to DNA DSB repair using the trapezoidal method based on the experimental data and the graphs obtained. This method evaluates the area under the curve reflecting changes in both γH2AX and Rad51 foci number [42]. The fraction of DNA DSB fixed by the HR pathway was determined by the area’s ratios under the Rad51 and γH2AX curves divided by the characteristic lifetimes’ ratios of the γH2AX and Rad51 foci during 24 h after irradiation. The proportion of γH2AX foci number, reflecting the fraction of DNA DSBs resolved by the HR pathway, can be interpreted as a “correctly” repaired DNA DSBs. The contribution of Rad51-linked DNA DSB repair of parental A549 and H1299 cells within 24 h after irradiation did not depend on their p53 status and amounted to 8% and 7% (*p* = 0.04 between parental A549 and H1299 cells) of the total number of γH2AX foci, respectively. In comparison, the contribution of Rad51-related DNA DSBs repair in p53null H1299IR cells (14%) was twice higher than that of p53wt A549IR cells (6%, *p* = 0.006).

### 2.3. Partial EMT-Program Activation in Parental and MFR-Surviving Cells

EMT represents a cellular program that confers on neoplastic epithelial cells the biological traits needed to accomplish most of the invasion–metastasis cascade [43].

Our previous results demonstrated that MFR-surviving population of p53-deficient cells had more prominent phenotypic signs of partial EMT-program activation (a spindled or rounded shape, loss of cell-cell contacts and loss of ability to anchorage-dependent colony formation, increased expression of ABCG2 protein—a marker of SP population of CSCs) in contrast with parental cells and cells carrying p53wt. That indirectly indicated the more prominent partial activation of the EMT program in p53-deficient MFR-surviving cells that is usually closely linked to entrance into the CSC state of different carcinoma cells [43].

Therefore, it was of interest to carry out a comparative analysis of the expression of E-cadherin, N-cadherin, and Vimentin, the marker proteins of EMT.

As evident from Figure 3, the p53wt MFR-surviving A549IR cells had almost 10-fold lower E-cadherin expression than their parental cells. The same MFR-surviving cells demonstrated a significant increase in mesenchymal marker N-cadherin, thus clearly indicating one of the partially activated EMT-program signs. While the p53null MFR-surviving H1299IR cells and their parental cells do not express E-cadherin endogenously, they both demonstrated lower N-cadherin expression compared to p53wt cells. This data suggested that other mesenchymal markers can be at play conferring EMT-program in cells in the absence of p53. Indeed, Vimentin’s expression, another mesenchymal marker, was almost twice as high in MFR-surviving p53null H1299IR cells compared to their parental cells and the p53wt cells (both MFR-surviving and parental) (Figure 3b). Of note, the extent of Vimentin cleavage in p53null parental and MFR-surviving cells was significantly greater than in p53wt (both IR and parental) cells.

### 2.4. Migration Activity

In lung cancers metastasis-related recurrence is still common and responsible for cancer-associated mortality. Previously, we observed the morphological changes of A549IR and H1299IR cells acquiring a spindled or rounded shape, loss of cell-cell contacts, increased number of viable round cells with reduced adherence to the plastic [38]. These results, together with observed biochemical signs (E-cadherin, N-cadherin and Vimentin expression) in H1299IR (p53-deficient) and A549IR (p53-wild type) cells suggested a partial activation of EMT process often associated with the highly invasive (migratory) phenotype of cancer cells. To determine whether 1D confined migration activity of the MFR-surviving cells is related to their p53 status, we performed wound healing (“scratch”) cell assay using parental and IR-surviving sublines.

There were no differences in the wound size exhibited by p53wt A549IR compared with the parental cells, implying that the decreased E-cadherin and increased N-cadherin expression observed in our study does not affect the 1D cell migration activity (Figure 4b). On the contrary, the p53null H1299 cells demonstrated higher wound healing compared to p53wt A549 cells. While the 1D migration activity of MFR-survived H1299IR cells attenuated at 24–48 h, it oddly reached their parental cells’ level already by 72 h. These data suggested that the absence of p53 significantly affected the 1D confined migratory behavior (wound healing) of NSCLC cells before and after MFR.

### 2.5. CD44high/CD166high (CSC-like) Population in Parental and MFR-Surviving Sublines after Additional Acute Irradiation

Well known that EMT-related signaling pathways contribute to tumor cell reprogramming into CSCs, metastatic tumor spread, and radioresistance [19,44,45,46,47]. In our previous study [38], MFR-surviving p53null subline of NSCLC cells demonstrated a significant decrease in the kinetics of IR Induced Nuclear Foci (IRIF) number (pATM и γH2AX) compared to parental cells during 8 h after single 2 Gy IR exposure. These cells did not show an apoptotic response, instead exhibited a dose-dependent decrease in apoptosis. That may indicate the involvement of anti-apoptotic regulatory mechanisms in MFR survivals. Whereas MFR-surviving p53wt cells, as expected, possessed a dose-dependent increase in apoptosis. These data suggested the presence of the CSC fraction, along with a possible change in their amount in the populations of MFR-surviving sublines.

Previously, the CD44high subpopulation of H1299 cells exhibited signs of EMT-program activation (including increased expression of CDH2 and VIM mRNAs) and a survival advantage under chronic cisplatin treatment [48]. Therefore, we carried out a quantitative assessment of the number of CD44high/CD166high cells, the fraction of CSC-like cells in untreated and IR-exposed parental and MFR-surviving p53wt vs. p53null sublines.

The CD44high/CD166high cells’ fraction was significantly enriched in the MFR-surviving subline compared to their parental p53wt cells after 24 h of cultivation without prior irradiation (Figure 5). On the contrary, in the absence of p53, the same cells’ proportion was almost equal in the MFR-surviving and parental H1299 cells cultivated under similar conditions. After acute single IR exposure to 2 Gy and 5 Gy, the proportions of CD44high/CD166high MFR-surviving cells were still significantly higher than their parental A549 cells, subtly increasing over parental non-irradiated cells only at dose 5 Gy. Thus, p53 status does not significantly affect the number of CSC-like cells in parental cells and their MFR-surviving sublines in response to single irradiation at doses 2 Gy and 5 Gy.

### 2.6. Rad51 Expression Indicates the Increased Activity of HR Pathway in CD44high/CD166high (CSC-Like) Population of Parental and MFR-Surviving Sublines

To assess the contribution of HR to DNA DSB repair, we analyzed the proportion of Rad51 positive (Rad51 +) cells in the population of FACS-enriched CD44high/CD166high cancer stem-like (CSC) and non-CSC (CD44low/CD166low) populations of MFR-surviving and their parental NSCLC cells. After 6 and 24 h of 2 Gy and 5 Gy of IR exposure, cells were trypsinized and analyzed by flow cytometry (Figure 6). The proportion of Rad51 + cells was calculated for each population (CSC and non-CSC) separately, taken as 100% (Figure 6).

As shown in Figure 6, the proportion of Rad51 + cells was significantly higher in the CD44high/CD166high population than in the CD44low/CD166low population in all groups with and without irradiation at any dose, thus, indicating the increased activity of the HR system in cancer stem-like cells. Notably, in the absence of p53, there was a significant increase in the proportion of Rad51 + cells in the CD44high/CD166high population of the MFR-surviving H1299IR subline over the parental H1299 cells at 6 h in contrast to the cell lines carrying p53wt (both A549 and A549IR).

### 2.7. DNA-PK(S) Indicates an Attenuated DNA DSBs Repair by NHEJ in p53-Deficient Cells

DNA-PKcs phosphorylation is critical for NHEJ mediated DSB repair [49]. DNA-PKcs phosphorylation at serine 2056 (S2056) cluster is an authentic autophosphorylation site in vivo and is essential for DNA-PK-mediated DSB repair [50]. To assess the contribution of NHEJ to DNA DSB repair in cancer stem-like populations of parental and MFR-surviving NSCLC sublines, we analyzed the proportion of DNA-PKcs (S2056) -positive (DNA-PK (S) +) cells irradiated at single doses of 2 Gy and 5 Gy. Cells were removed from the plastic surface by trypsinization at 6 h and 24 h after irradiation and analyzed by flow cytometry (Figure 7). The proportion of DNA-PK(S) + cells was calculated for each population (CSC and non-CSC) separately, taken as 100%.

The proportion of DNA-PK(S) + cells was higher in CD44high/CD166high populations of all studied groups. In the presence of p53wt, the proportion of DNA-PK(S) + cells in CD44high/CD166high populations of A549 and A549IR subline increase 24 h after irradiation as compared to 6 h (Figure 7a). In contrast, in the absence of p53, the proportion of DNA-PK (S) + cells in the same populations of H1299 и H1299IR sublines decreases from 6 h to 24 h after acute single dose IR exposure (Figure 7b).

Notable, while p53-deficient CSC-like fraction of both parental and MFR-surviving cells demonstrated IR dose-dependent increase in the proportion of DNA-PK(S) + cells at an early time (6 h) point after irradiation, the p53-competent possess such a response neither at early nor late time points. This effect might indicate that p53wt ensures DNA-PK-mediated DSB repair for both parental and MFR-surviving cells irrespectively of a subsequent acute single dose of their IR exposure. Whereas in the absence of p53, a dose-dependent increase of DNA-PK-mediated NHEJ occurs as an early post-irradiation acute response is more intensive in the CSC-like population MFR-surviving H1299IR cells, compared to their parental H1299 cells. Nevertheless, these p53null cells’ response of somehow declined later, indicating a slower DNA DSBs repair by this mechanism.

### 2.8. LC3 Activity Confers Cancer Stem-Like Cells Undergo Autophagy Process after Irradiation Exposure

Autophagy is thought to play a crucial role in cancer cells’ resistance to various anticancer therapies [51]. As an essential transfection factor, p53 promotes autophagy by transactivating its target genes that involved the diversity of radiation responsive pathways in mammalian cells [52]. Besides, many publications provided experimental evidence that the pro-oncogenesis and metastatic activity of autophagy may be attained by augmenting the stemness of CSCs [53,54,55]. Nonetheless, the impact of p53-related autophagy in radiosensitivity of MFR-surviving cells and their CSC-like subpopulations remains unclear and debatable.

To address the involvement of autophagy and p53 status in response to radiation, we analyzed the extent of autophagic markers LC3-II/LC3-I in the populations of parental and MFR-surviving cells. A lipidated form of cytosolic protein LC3-I, LC3-II, is considered an autophagosomal marker in mammals [56]. Therefore, we evaluated both these marker proteins in FACS-enriched CD44low/CD166low and CD44high/CD166high populations of parental and MFR-surviving NSCLC cells 24 h after additional X-ray exposure at doses of 2 Gy and 5 Gy (Figure 8). We used the antibody detecting both the non-lipidated LC3-I (non-autophagic) and lipidated LC3-II (autophagic) under autophagy conditions to estimate the proportion of LC3-positive (LC3 + ) cells.

The CD44high/CD166high cells demonstrated a significantly higher proportion LC3 + cells than CD44low/CD166low populations irrespectively of their p53 status, indicating the critical role of autophagy in the survival of CSC-like cells. Of note, CD44high/CD166high populations of parental A549 and H1299 cells demonstrated a dose-dependent increase in the percentage of LC3 + cells after 2 Gy and 5 Gy irradiation compared to nonirradiated controls (Figure 8a,b). However, for A549 cells statistically significant difference was observed only after 5 Gy irradiation. Whereas the autophagy activity in CD44high/CD166high populations of the A549IR and H1299IR sublines did not change after additional single-dose irradiation.

## 3. Discussion

Epithelial and mesenchymal markers are the most intensively studied for putative therapeutic targets of CSCs in solid tumors. As epithelial cancer cells undergo EMT, they gain stemness, motility, invasiveness, drug resistance, and angiogenic and metastatic ability [7]. EMT transition in epithelial cells leads to switching from E-cadherin to N- cadherin and increasing Vimentin expression [7]. CD133, CD44, ALDH1A1, CD90, and CD166 were verified to be effectively used as CSC markers in NSCLC [57,58]. Epithelial tumor cells with ongoing EMT develop CSC traits. Our study confirms that MFR promotes EMT marker N-cadherin expression and downregulates E-cadherin expression in p53wt cells. We also demonstrate the predominant upregulation of Vimentin’s expression in H1299IR compared to their parental and p53wt cells (Figure 3). Vimentin’s high expression was associated with tumor progression and metastasis [59,60]. The EMT process could be controlled by a ZEB1/mir200 feedback loop mechanism (maintaining stemness) and by p53 [61]. Loss of p53 function related to cells losing their epithelial differentiation and undergo EMT, characterized by high expression of Vimentin and Snai1, Snai2, Twist, ZEB1 and ZEB2, the EMT inducers [62,63]. However, the observed spindle-shaped and rounded cell morphology of H1299IR cells was different from the typical lamellipodia morphology associated with high Vimentin [64], explaining the observed significant delay in their 1D migration activity (Figure 4) at 24–48 h compared to their parental cells. That could be due to the substantial decrease of the adherence of these cells, which, in turn, can explicate the difference between anchorage-independent and anchorage-dependent colony formation results (Figure 1). Attractive, the acute single-dose 2 Gy and 5 Gy irradiation induced reduction of migration and the increased adhesion of A549 cells within 24 h after irradiation [65]. Previously, we observed a reduction of MFR-survived A549IR cells’ adhesion, albeit without any changes in their migration using the same type of migration assay in our present study (Figure 4). Such an apparent discrepancy can emphasize the significant difference in tumor cellular response to acute and multifractionated IR exposure regarding cellular migratory and adhesive properties. In this regard, our present data corroborate previous reports on increased motility of H1299 human NSCLC cells [66], supplementing other contradictory reports on the effects of IR on cancer cell migration [67,68,69]. In sum, p53 status significantly affected the 1D confined migratory behavior (wound healing) and an irradiated cell’s capability to continue to divide and form a colony of NSCLC cells before and after MFR.

Two main pathways involved repairing DNA DSBs in the mammalian cell: canonical NHEJ and HR pathway. Canonical NHEJ is the predominant repair pathway in all mammalian cell cycle phases and is used by the cells to repair most DSBs. HR contributes to repairing DNA DSBs only in S and G2 phases when sister chromatids can be used as a template to repair the damage [70]. These two pathways make a different contribution to the two-component kinetics of DNA DSBs repair. The rapid phase (2–4 h) uses canonical NHEJ and is estimated to repair ~40–60% of DSBs. Simultaneously, the slow kinetics occurs in S and G2 phases of the cell cycle by the HR pathway. DNA damage caused by ionizing radiation leads to phosphorylation of H2AX that is mediated by all PIKK kinases, ATM, ATR, and DNA-PK [71,72]. The phosphorylated H2AX (γH2AX) foci are regarded as recruiting mediators of the repair factors of DNA DSBs [73,74]. The H2AX is one of the essential components of the active ATM complex and participates in the activation of ATM-dependent phosphorylation of cell cycle checkpoint-related factors such as p53, Chk2, SMC1, and NBS1 [75]. The rate of γH2AX decay and relative residual damage after exposure to IR might be useful as indicators of intrinsic radiosensitivity, the more rapid loss and less retention appeared in the more radioresistant cell types [76]. In this regard, cervical cancer cells with wild-type p53 showed a significantly faster γH2AX decay rate after irradiation than cells deficient in p53 [77]. Here, we also found that the level of intrinsic and remaining foci at 24 h after 2 Gy IR exposure was higher in p53null cells than in p53wt cells. Notable, the MFR-surviving A549IR cells at 8 h and 24 h also had less γH2AX foci than H1299IR, which had more rapid and significant γH2AX decay compared to their parental cells (Figure 2b,d). Although we cannot exclude that after 24 h, further decay of the γH2AX foci might also occur in the p53 null cells, whether these residual DSBs correlated with radiosensitivity remains obscure [73]. Thus, our data further emphasize the impact of p53 status on the decay of γH2AX foci and associated efficacy of the DSB repair in NSCLC cells survived after MFR, prompting further investigation of its role in their radiosensitivity.

Known that the p53-p21 system defect leads to an increase in the proportion of H1299 cells in the S phase due to the formation of DNA DSB and an internal S-block, leading to an increase in the S phase length [78]. In our previous study, irradiation at doses of 2–6 Gy increased the proportion of cells in the S and G2/M phases in the H1299IR (p53-null) compared to the A549IR (p53wt) subline [38]. DNA repair protein Rad51 homolog 1 (Rad51) is an essential protein in the HR repair pathway. In the present study, the contribution of Rad51 to DNA DSB repair in the H1299IR subline was the highest among the four cell lines studied (Figure 2c,e). Thus, the delayed kinetics of DNA DSB repair in H1299IR cells may be associated with *de novo* DSB formation during the S phase and with slow DNA DSB repair due to HR in the absence of p53.

Indeed, p53 may participate in the regulation of Rad51. Reintroduction of p53wt into p53 mutant soft tissue sarcoma (STS) cell lines resulted in decreased Rad51 protein and mRNA expression. [79]. Moreover, resulting from the DNA damage caused by IR, the p53/p63/p73 family of transcription factors can target Rad51 [80]. The p73 isoforms are known to exhibit similar p53 transcriptional activity in H1299 cells (p53null) [81,82].

One explanation for tumor cells’ resistance to radiotherapy is CSCs in a heterogeneous tumor structure. Several studies suggest that Rad51 may play a significant role in CSC’s resistance to therapy [20,29,83]. CSCs are proposed to repair DNA damage more efficiently than the rest of tumor cells [84], and can be carried out by the HR mechanism [29,30,31,32]. Therefore, the high proportion of Rad51 + cells in the CSC population of H1299IR cells (p53null) compared to A549IR (p53wt) may be associated with delayed DNA DSB repair processes, but also with the high levels of Rad51 expression due to the activity of other p53 family proteins. Our present data cannot exclude that in the absence of p53wt, the other members of the p53 family could ensure more efficient DNA DSB repair by the HR mechanism, leading to resistance of p53-deficient tumor cells to chemo- and radiotherapy. In this regard, we previously found a high content of the EMT-like phenotype cells, somewhat similar to the SP phenotype in the bulk of our MFR-surviving sublines [38]. Indeed, more effective HR in such fractions may provide a high survival, pointing to a significant role of p53-family proteins (in particular, p73 and/or p63) in conferring a stem-like cell and radioresistant phenotype of NSCLC cells that is associated with our previous observation for the ABCG2-overexpressing H1299IR cells. However, this hypothesis warrants further investigation in our ongoing studies.

CD44, a cell-surface extracellular matrix receptor, is commonly used as CSC markers [58]. Known functions of CD44 are cellular adhesion (aggregation and migration), hyaluronate degradation, lymphocyte activation, lymph node homing, myelopoiesis and lymphopoiesis, angiogenesis, and release of cytokines. The physiological functions of CD44 indicate that the molecule could be involved in the metastatic spread of tumors [85] and reported among human lung CSC markers [7]. CD166, also known as activated leukocyte cell adhesion molecule (ALCAM), has been identified as a cell surface marker of specific hematopoietic progenitor and pluripotent mesenchymal stem cells and is frequently considered a colorectal CSC marker [58]. We confirm a dose-dependent increase in CD44 and CD166 expression levels in A549IR, but not in H1299IR cells 24 h after irradiation compared to their parental cells (Figure 5). Therefore, status p53 may influence the CD44/CD166 expression level in MFR-surviving NSCLC cells after an acute dose of IR, contributing to a high risk for metastasis with anticipated low patient survival.

The DNA-dependent protein kinase (DNA-PK), a serine/threonine protein kinase, consists of a catalytic subunit (DNA-PKcs) and a Ku heterodimer, building up by the Ku70 and Ku80 subunits. The choice between NHEJ and HR pathway attributed to the DNA-PK’s role in the DDR pathways [86]. The DNA-PKcs activation after exposure to IR results in rapid phosphorylation at the T2609 phosphorylation cluster and S2056 residue [87,88]. DNA-PKcs phosphorylation at the T2609 cluster is primarily mediated by ATM kinase, whereas DNA-PKcs phosphorylated at S2056 residue is an authentic autophosphorylation site in vivo [50]. In the context of DNA repair pathway choice between HR and NHEJ, BRCA1 prevents NHEJ in the S and G2 phases by inhibiting DNA-PKcs autophosphorylation at S2056 [89]. Highly likely, the phosphorylation-mediated activation of DNA-PKcs makes DNA repair pathway choice: a phosphorylated/active form of DNA-PKcs errands NHEJ, while an unphosphorylated/inactive form favors HR. Nonetheless, the other factors persuading DDR pathway choice remain obscure since the mutational inactivation and small-molecule inhibition of DNA-PKcs favored and compromised HR, respectively [90]. In this regard, it is worthy of mentioning the interplay of replication protein A (RPA), p53, and Rad51. RPA is an essential HR modulator, being a heterodimer that binds to single-stranded DNA (ssDNA). DNA-PKcs hyperphosphorylated RPA in complexes with p53 after DNA damage. With the phosphorylation of p53, this hyperphosphorylation leads to decay of the RPA-phospho-p53 complex allowing RPA to bind to ssDNA and promote HR via Rad51. The picture becomes even more complicated, where the expression of catalytically inactive DNA-PK causes more genomic instability than the loss of the proteins themselves. Being unique for each kinase, the spectrum of genomic instabilities and physiological consequences depends on their activating complexes. That allowed suggesting a model in which the catalysis coupled with DNA/chromatin release and catalytic inhibition leads to the persistence of the kinases at the DNA lesion, which, in turn, affects repair pathway choice and outcomes [91]. In the present study, we focused on evaluating a phosphorylated/active form of DNA-PKcs (DNA-PK (S) +) related to p53 status in MFR-surviving and parental NSCLC cells.

Before following any DNA repair pathway, a cell needs to detect the presence of DSBs. The H2AX is phosphorylated both by DNA-PKcs and ATM at its Ser139 residue to form γH2AX, a marker of DNA damage, that functions to retain factors involved in DSB repair [92]. Here, we observed the γH2AX foci decay by only 65% at 8 h after irradiation (Figure 2d) in the bulk of p53null cells, and the acute (6 h) dose-dependent rise with subsequent (24 h) decline of the proportion of DNA-PK (S) + cells in the CD44high/CD166high (CSC-like) population of MFR-surviving cells (Figure 7b). That could highlight that delayed DNA damage repair by the NHEJ mechanism not occurred in the entire population. Somewhat more likely, the delayed NHEJ mechanism-based repair appears mainly in the CSC-like cells not carrying p53wt.

Moreover, in the H1299IR subline, HR’s contribution to DNA DSB repair was the highest among all four studied NSCLC cell populations (14%).Simultaneously, a high proportion of Rad51 + cells indicates increased HR activity in CSC-like cells (Figure 6). Similar to our present data, the irradiated glioblastoma CSCs (GBM CSCs) had a significant delay in the phosphorylation of H2AX and DNA-PKcs (phospho S2056) up to 72 h and 48 h after irradiation, respectively [93]. Besides, the DNA DSB repair delay along the NHEJ pathway in these cells was observed until 48 h after irradiation. These data are consistent with our present results for parental and MFR-surviving cells devoid of p53wt.

Comparing our data in Figure 6 and Figure 7, the presence of functional p53 did not lead to any significant changes in DNA DSB repair by either the HR or NHEJ mechanism in the CSC-like population of parental A549 cells in response to acute X-ray irradiation at the single-dose of 2 Gy and 5 Gy. The CSC-like population of A549IR cells had the same effect, although the higher level of DNA DSB repair by the NHEJ mechanism in unirradiated and irradiated cells compared to their parental cells 24 h after irradiation. An increase in the proportion of Rad51 + and DNA-PK (S) + cells 24 h after irradiation may reflect the fate of residual DNA DSBs. Besides, a high number of residual DNA DSBs may indicate unrepaired DSBs and genetic instability processes.

In the absence of p53, NHEJ makes the main dose-dependent contribution to repairing DNA DSB in the CD44high/CD166high population of parental H1299 cells and only at the early time point (6 h) of response to acute single X-ray irradiation. Both NHEJ and HR contribute to the DNA DSB repair in H1299IR cells at the early stages (6 h). Besides, the overall level of DNA DSB repair by both mechanisms in H1299 IR cells was significantly higher (2–5 times) than in parental cells before and after additional single-dose irradiation. Thus, in response to additional irradiation, the p53wt favors neither HR nor NHEJ mechanisms of sustained DNA DSB repair in both MFR-surviving and parental NSCLC cells. Whereas in the absence of p53, the other p53 family proteins likely confer mainly early but not sustained DNA repair by both mechanisms.

Autophagy plays a role in maintaining stem cells in normal stem cells and CSCs [53,94,95]. Exposure to IR leads to an increase in autophagy marker protein expression in CSCs [96]. Moreover, there is now increasing evidence that autophagy signaling and EMT are interlinked, and the autophagy is often highly expressed in tumor cells carrying a mesenchymal signature [36]. Microtubule-associated protein light chain 3 (LC3) is the mammalian homolog of yeast Atg8), which is involved in the elongation and closure of the autophagosomal membrane [36]. LC3 belongs to a novel ubiquitin-like protein family involved in different intracellular trafficking processes, including autophagy [97]. Generated by the conjugation of cytosolic LC3-I to phosphatidylethanolamine (PE) on nascent autophagosomes, the LC3-II is considered their standard marker. In our study, the p53wt cells demonstrated the significant dose-dependent autophagic response in the CD44high/CD166high populations of both parental and isogenic MFR-surviving cells after exposure to a single acute IR dose (Figure 8a,b). The p53null parental cells also possessed a similar response, whereas the isogenic MFR-surviving cells instead demonstrated the trend for a decline under similar IR exposure conditions. However, our data should be taken with caution. Since LC3-II, but not LC3-I, is required to form an autophagosomal membrane, we cannot speculate definitely about autophagy activity in our cells. Nevertheless, our results of autophagy in CD44 high/CD166 high cells are quite impressive, especially in the light of very recent data published while our paper was under review. In this paper, the authors demonstrated three increased autophagy markers (LC3 and P62 by WB; RFP-LC3 puncta; TEM analysis of the cytoplasmic accumulation of autophagosomes) in the A549-oncosphere, but less in the H1299-oncosphere CSC cells, indicating that the autophagic activity is higher than that of in the respective parental cell lines [98]. Although the study is based on an isolated CSC population, these data corroborate our current findings entirely on the autophagic activity of the CD44 high/CD166 high population of MFR-survived cells. Our study also strictly observed a significantly higher content of LC3 + cells in the CD44 high/CD166 high populations of p53wt MFR-surviving cells (Figure 8), which enriched the CSC-like cells (Figure 5) in contrast to their p53null counterparts. Therefore, our autophagy data sounds very relevant, indicating the higher intrinsic and IR-induced autophagic activity of MFR-surviving p53wt over p53null cells. Interestingly, the authors postulated that autophagy augments the stemness only in A549-oncosphere cells, but not the H1299-oncosphere cells via degradation of ubiquitinated p53wt, providing the possible mechanism underlying our observations.

However, another uncertainty is whether radioresistance increased in association with the promotion of autophagy. Our previous [38] and current (Figure 8) data indicated that both apoptosis and autophagy dose-dependently increased in the bulk of parental and MFR-surviving p53wt cells in response to acute IR exposure. On the contrary, the bulk of p53null cells possessed the decline of apoptosis and autophagy under the same conditions. Our experimental design could not exclude that other modes of cell death (such as mitotic catastrophe or senescence) might be engaged, responding to a single acute dose of X-ray in IR-surviving cells in the absence of p53. The study mentioned above may also provide a possible molecular “switch” of nonprotective autophagy in p53wt NSCLC cells to protective autophagy in our MFR-surviving cells.

Of clinical importance, autophagy can have both pro-and anti-tumorigenic effects. The regulation of the self-restoration of the NSCLC CSCs could be achieved through p53-ZEB1 axis, contributing to the recently suggested new CSC-based mechanism that underlies autophagy’s oncogenic activity [98]. On the other hand, some recent data indicated that radioresistance appeared to be most closely associated with senescence occurring earlier and more significant in the p53wt NSCLC cells [99]. In the light of a tumor dormancy model, a phase of proliferative recovery in a subset of cells frequently following after the radiation-induced senescence can contribute to therapy resistance and conceivably the disease relapse [100,101]. In this regard, such an impact of senescence will open up a new avenue for novel senolytics that await clinical prove as an adjuvant to radiotherapy to prolong remission by delaying disease recurrence in patients. While being out of our current study’s scope, these findings prompted us to investigate further the impact of senescence on radiosensitivity of our MFR-surviving NSCLC sublines concerning their p53 status in our forthcoming studies.

In the end, our data emphasize that careful assessment of other vital processes (such as senescence and autophagy-p53-Zeb1 axis in CSCs) of primary radiation responses may provide new potential targets modulated for therapeutic benefit radiosensitizing cancer cells while rescuing normal tissue. Besides, our migration and autophagy data can shed light on the sophisticated crosstalk between autophagy and the p53-related EMT, by which cancer cells obtain an invasive phenotype and metastatic potential.

## 4. Materials and Methods

### 4.1. Cell Culture

A549 cell line was obtained from ATCC (Manassas, VA, USA) and cultured in DMEM (Gibco, Thermo Fisher Scientific, Waltham, MA, USA) containing 10% FBS, L-glutamine, and 1% penicillin/streptomycin (Sigma-Aldrich, St. Louis, MO, USA). H1299 cell line was obtained from ATCC (Manassas, VA, USA) and cultured in RPMI-1640 (Gibco, Fisher Scientific, Waltham, MA, USA) containing 10% FBS, L-glutamine, and 1% penicillin/streptomycin (Sigma-Aldrich, St. Louis, MO, USA). Cells were maintained in a humidified 5% CO_2_ environment at 37 ͦC.

### 4.2. Irradiation

Exponentially growing A549 and H1299 cells were irradiated using 200 kV X-ray RUB RUST-M1 X-irradiator (JSC “Ruselectronics”, Moscow, Russia) at the dose rate of 0.85 Gy/min (2.5 mA, 1.5 mm Al filter) at room temperature. A total dose of 60 Gy was divided into several doses as follows: 2 Gy of 10 fractions, 5 Gy of 4 fractions, and 10 Gy of 2 fractions. Cells were incubated for up to 3–4 days between fractionated doses of 5 Gy and 10 Gy as a recovery period. After the last exposure, cells were maintained in normal growth conditions for 3 weeks to recover.

### 4.3. Colony Formation and Anchorage-Independent Soft Agar Assay

Exponentially growing A549, A549IR, H1299 and H1299IR cells were irradiated at single dose irradiation of 0 Gy, 2 Gy, 4 Gy, and 6 Gy. Immediately after irradiation cells were seeded on 60 mm Petri dishes at density of 150, 500, 1000 and 2000 cells/well for each radiation dose. After 2 weeks, cells were fixed with methanol for 15 min and were stained with Giemsa for 15 min. Plating efficiency (PE) and survival fraction (SF) were calculated using the following equations:PE = # of colonies formed # of cells seeded × 100% (1)
SF = # of colonies formed after irradiation/[# of cells seeded × PE](2)

Anchorage-independent soft agar assay was performed as described elsewhere [102]. Briefly, 6-well plates were coated with 1.0% noble agar in complete media (1.5 mL agar/well) and allowed it to solidify at room temperature for 30 min. Immediately after irradiation cells were trypsinized. Cell suspensions were mixed with 0.6% noble agar, added to wells and allowed to solidify for another 30 min. After 21 days colonies were stained with 0.05% Crystal Violet for 1 h and washed with PBS. The number of colonies was calculated manually.

### 4.4. Immunofluorescence Staining

The Immunofluorescence Staining protocol was performed as previously described [103,104]. Cells were seeded at the density of 6 × 10^3^ cells/cm^2^ onto coverslips (SPL Lifesciences, Gyeonggi-do, South Korea) placed inside 35 mm Petri dishes (Corning, New York, NY, USA) 48 h before irradiation and fixed 0.5–24 h after irradiation in 4% paraformaldehyde in PBS (pH 7.4) for 15 min at room temperature, followed by two rinses in PBS and permeabilization in 0.3% Triton-X 100 (in PBS, pH 7.4) supplemented with 2% bovine serum albumin (BSA) to block non-specific antibody binding. Cells were then incubated for 1 h at room temperature with primary mouse monoclonal antibody against γH2AX (dilution 1:200, clone JBW 301, Cat. # 05-636, Merck-Millipore, Burlington, VT, USA), primary rabbit polyclonal antibody against Rad51 (dilution 1:200, Cat. # ABE257, Merck Millipore, Burlington, VT, USA), which were diluted in PBS with 1% BSA and 0.3% Triton-X 100. After several rinses with PBS, cells were incubated for 1 h with secondary antibodies IgG (H + L) goat anti-mouse (Alexa Fluor 555 conjugated, dilution 1:800; Cat. # A-21424, Merck-Millipore, Burlington, VT, USA), goat anti-rabbit (Alexa Fluor 488 conjugated, dilution 1:500; Cat. # A-11008, Merck Millipore, Burlington, VT, USA) diluted in PBS (pH 7.4) with 1% BSA. Coverslips were then rinsed several times with PBS and mounted on microscope slides with ProLong Gold medium (Life Technologies, Carlsbad, SA, USA) with DAPI. Cells in coverslips were imaged using Nikon Eclipse Ni-U microscope (Nikon, Tokyo, Japan) equipped with a high-definition camera ProgResMFcool (Jenoptik AG, Jena, Germany). Filter sets used were UV-2E/C (340–380 nm excitation and 435–485 nm emission), B-2E/C (465–495 nm excitation and 515–555 nm emission), and Y-2E/C (540–580 nm excitation and 600–660 nm emission). A total of 300–400 cells were imaged for each data point. Foci were counted by manual scoring.

### 4.5. Flow Cytometry Analysis

Exponentially growing cells were harvested by trypsin 1 × 10^6^ cells per sample, washed in PBS and incubated with primary rabbit polyclonal antibody against CD166 (dilution 1:200 Sigma-Aldrich, Saint Louis, MO, USA) and with secondary antibodies IgG (H + L) goat anti-rabbit (Rhodamin conjugated, dilution 1:300; Merck Millipore, Burlington, VT, USA) and mouse monoclonal antibody against CD44 (PE conjugated, clone MEM-263, dilution 1:70, Sigma-Aldrich, Saint Louis, MO, USA) for 1 h at 4 ᵒC. Cells were then fixed in 4% paraformaldehyde for 15 min at room temperature and permeabilized in 0.3% Triton-X 100 (in PBS, pH 7.4) supplemented with 2% bovine serum albumin (BSA) to block non-specific antibody binding. Cells were then incubated overnight at 4 ᵒC with primary rabbit polyclonal antibody against Rad51 (dilution 1:200, Merck Millipore, Burlington, VT, USA) and primary rabbit polyclonal antibody against DNA PKcs (phospho S2056) (dilution 1:200, Abcam, Cambridge, USA) or primary rabbit polyclonal antibody against LC3-I/II (dilution 1:200, Merck Millipore, Burlington, VT, USA). After several rinses with PBS cells were incubated for 1 h with secondary antibodies IgG (H + L) goat anti-rabbit (Alexa Fluor^®^ 488 conjugated, dilution 1:500; Merck Millipore, Burlington, VT, USA). Cells were analyzed by flow cytometry (BD FACSCalibur, Becton Dickinson, San Jose, CA, USA). A total of 50,000 events were acquired for each sample and the proportion of positive cells was analyzed with BD CellQuest Pro 5.1 software (Becton Dickinson, San Jose, CA, USA).

### 4.6. Western Blotting

After incubation, cells cultured in a T75 flask were collected and washed twice with cold PBS and lysed in 1 mL of a lysis buffer (150 mM sodium chloride, 1.0% Triton X-100, 0.5% sodium deoxycholate, 0.1% SDS (sodium dodecyl sulfate), 50 mM Tris–HCl (pH 8.0)) and centrifuged at 14,000× *g* for 25 min at 4 ᵒC to yield whole-cell lysates. Protein concentration was measured using the bicinchoninic acid (BCA) method (Thermo Scientific™ Pierce™ BCA Protein Assay Kit, Rockford, IL61105 USA). Aliquots of the lysates (30 μg of protein) were separated on a 8–16% SDS–polyacrylamide gel (Bio-Rad Laboratories, Mini-PROTEAN TGX Gels, Hercules, CA, USA) with a 10× Running Buffer (glycine transfer buffer [192 mM glycine, 25 mM Tris–HCl (pH 8.3), 1% SDS (*v*/*v*)] and transferred onto Mini-size nitrocellulose membranes (7.1 × 8.5 cm, Bio-Rad Laboratories, Trans-Blot^®^ Turbo™ Transfer, Neuberg, Germany) using Mini-size Transfer Stacks (7.1 × 8.5 cm, Bio-Rad Laboratories, Trans-Blot^®^ Turbo™ Transfer, USA) and 5× Transfer Buffer (Bio-Rad Laboratories, Trans-Blot^®^ Turbo™ Transfer, USA). After blocking the non-specific site with Blocking Buffer (Thermo Scientific™ Pierce™ Protein-Free (PBS) Blocking Buffer, Waltham, MA, USA), the membrane was incubated with primary polyclonal Rabbit Anti-Vimentin antibody—Cytoskeleton Marker (1 µg/mL, ab45939, Abcam, Cambridge, MA, USA), Anti-E-Cadherin antibody [EP700Y]—Intercellular Junction (dilution 1:1000, ab40772, Abcam, Cambridge, MA, USA) and Anti-N- Cadherin antibody [5D5]—Intercellular Junction Marker (dilution 1:2000, ab98952, Abcam, Cambridge, MA, USA) in Blocking Buffer at 4 °C for overnight. Membrane was washed with 1× PBS (pH = 7.2–7.6), (Eco service, St. Petersburg, Russia) containing 0.05% Tween-20 (Pharm grade, Biolot, St. Petersburg, Russia) 3 times for 3 min each. The membrane was further incubated for 2 h with a Peroxidase conjugated secondary sheep anti rabbit (p-SAR IgGs) (dilution 1:5000, IMTEC, Moscow, Russia) and sheep anti mouse (p-SAM IgGs) (dilution 1:1000, IMTEC Ltd., Moscow, Russia) antibodies at room temperature. Membrane was washed with TPBS (1× PBS containing 0.05% Tween-20) 5–8 times for 5 min each. The formed membrane-bound immune complexes were detected using the Clarity™ Western ECL Substrate reagent Luminol/peroxide solution (dilution 1:1, Bio-Rad, Hercules, CA, USA). Normalization was done staining the total protein with Ponceau S (0.1% Ponceau S and 5% acetic acid). Blots were visualized, and the relative densities of bands were calculated by ChemiDoc™ MP Imaging System (170-8280) by Bio-Rad (Hercules, CA, USA).

### 4.7. Migration Assay

A549 and H1299 cells were seeded at the density of 20 × 10^3^ cells per well in a 96-well plate, incubated at 37 °C, 5% CO_2_ and allowed to grow for 72 h. Once the cells reached 100% confluence, a straight-edged scratch was created using a sterile plastic micropipette tip to simulate an in vitro wound, cell-free zone across the cell monolayer in each well. After that, the monolayer was washed with PBS (pH 7.4) 3 times to remove cell debris, then complete medium was added. Images at time zero (t = 0 h) were captured to record the initial area of the wounds, and the recovery of the wounded monolayers due to cell migration toward the denuded area was evaluated at 24 h, 48 h and 72 h (t = Δ h). The images were captured using fluorescence microscopy (ImageXpress Micro XL, Molecular Devices LLC, San Jose, CA, USA). The area of wound was quantified using Custom Module Editor of MetaXpress 5.0 Software. The migration of cells toward the wounds was expressed as percentage of wound closure:% of wound closure = [(at = 0 h − at = ∆ h)/at = 0 h] × 100%,(3)
where, at = 0 h is the area of wound measured immediately after scratching, and a t = Δ h is the area of wound measured at 24 h, 48 h or 72 h after scratching.

### 4.8. Statistical Analysis

Statistical and mathematical analyses of the data were conducted using the Statistica 8.0 software (StatSoft) and EXCEL 2010 Software. The results are presented as means of three independent experiments ± standard deviation until otherwise stated. Statistical significance was tested using the Student *t*-test and Mann-Whitney U Test.

## 5. Conclusions

Our data indicate that the p53 status significantly affects the recognition and repair of DNA DSBs in CSC-like populations of NSCLC sublines, which may play an essential role in CSCs survival after MFR. Accurate assessment of the p53 status-related migration, autophagy, senescence, and EMT processes in CD44high/CD166high populations of MFR-surviving cells can reveal new potential targets of NSCLC cell resistance to radiotherapy, opening up an avenue for the development of new modulators radiosensitizing tumor cells while rescuing normal tissue for patients’ therapeutic benefit.

## Figures and Tables

**Figure 1 ijms-22-02369-f001:**
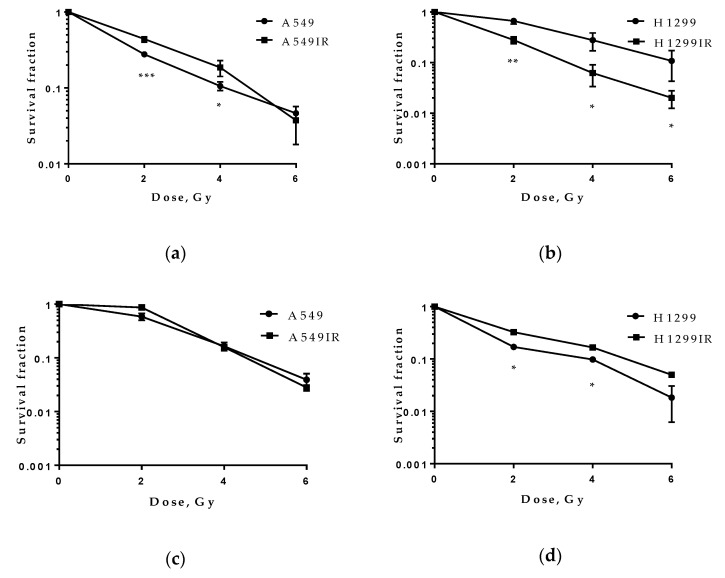
Radiosensitivity of A549 and A549IR cells and H1299 and H1299IR cells after X-ray exposures evaluated by colony formation assay (**a**,**b**) and soft agar assay (anchorage-independent growth) (**c**,**d**). * *p* < 0.05, ** *p* < 0.01; *** *p* < 0.001. Data are means ± SD of three independent experiments.

**Figure 2 ijms-22-02369-f002:**
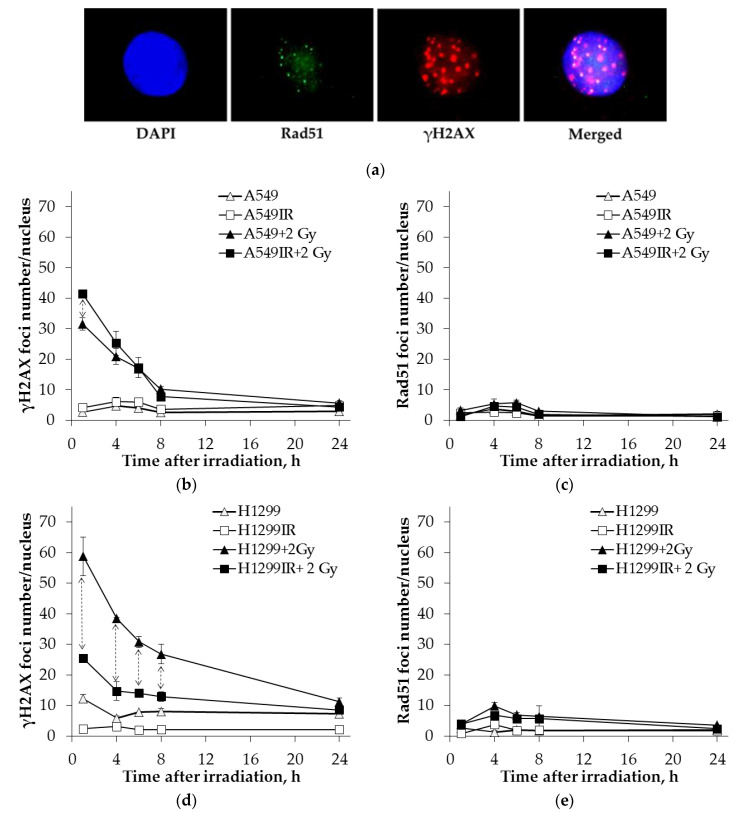
Kinetics of γH2AX and Rad51 foci changes in A549 and A549IR cells and H1299 and H1299IR cells after 2 Gy X-ray exposure. Representative immunofluorescent images of the irradiated cells showing Rad51 (green), γH2AX (red) foci and their colocolization (Merged). DAPI nuclear counterstaining is shown in blue (**a**). Comparative analysis of changes in the number of γH2AX foci in A549 and A549IR (**b**) and H1299 and H1299IR cells (**d**) after 2 Gy X-ray exposure; changes in the number of Rad51 foci in A549 and A549IR cells (**c**) and H1299 and H1299IR cells (**e**) after 2 Gy X-ray exposure. ↕ denotes significant differences between groups at *p* < 0.05. Data are means ± SD of three independent experiments.

**Figure 3 ijms-22-02369-f003:**
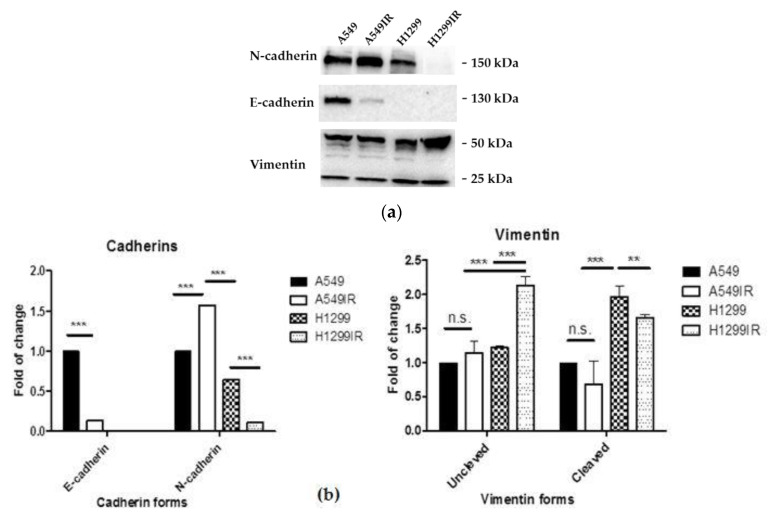
Expression of EMT-related markers in parental and MFR-surviving NSCLC sublines. Western Blot (**a**) and quantification (**b**) analysis of E-cadherin, N-cadherin and Vimentin expression. *** *p* < 0.001; ** *p* < 0.05; n.s. non-significant. Data are means ± SD of three independent experiments.

**Figure 4 ijms-22-02369-f004:**
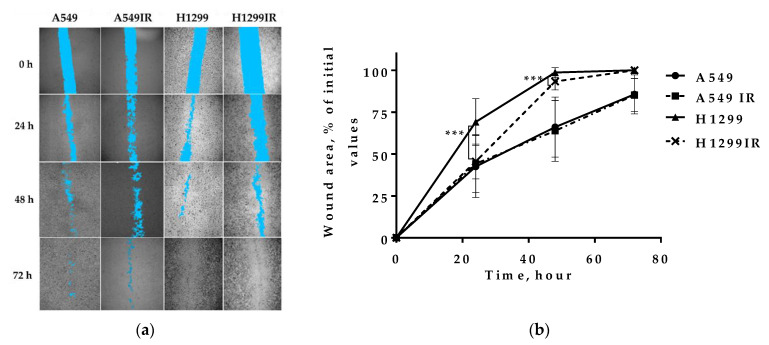
The wound healing assay was used to detect the migration ability of parental and MFR-surviving A549 and H1299 cells. (**a**) After mechanical wounding cells were grown for 24 h, 48 h and 72 h. Cell free area shown in blue. (**b**) Percentage of cell free area in each condition was calculated. *** *p* < 0.001.

**Figure 5 ijms-22-02369-f005:**
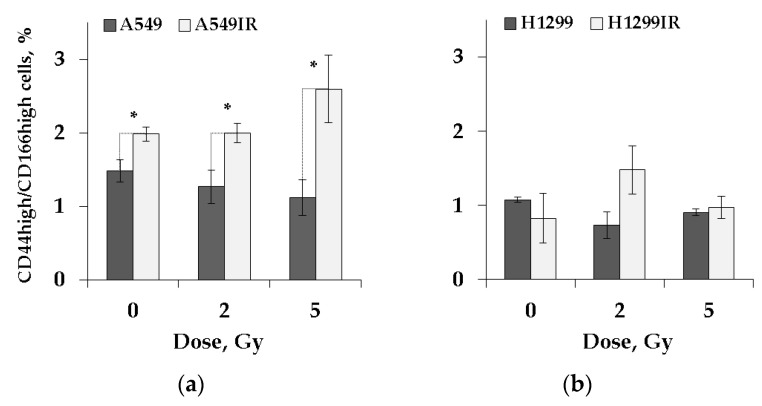
Evaluation of CD44high/CD166high (CSC-like) population in parental and MFR-surviving sublines. The proportion of CD44high/CD166high population in A549 and A549IR cells (**a**) and H1299 and H1299IR cells (**b**). Data are means ± SD of three independent experiments. * *p* < 0.05.

**Figure 6 ijms-22-02369-f006:**
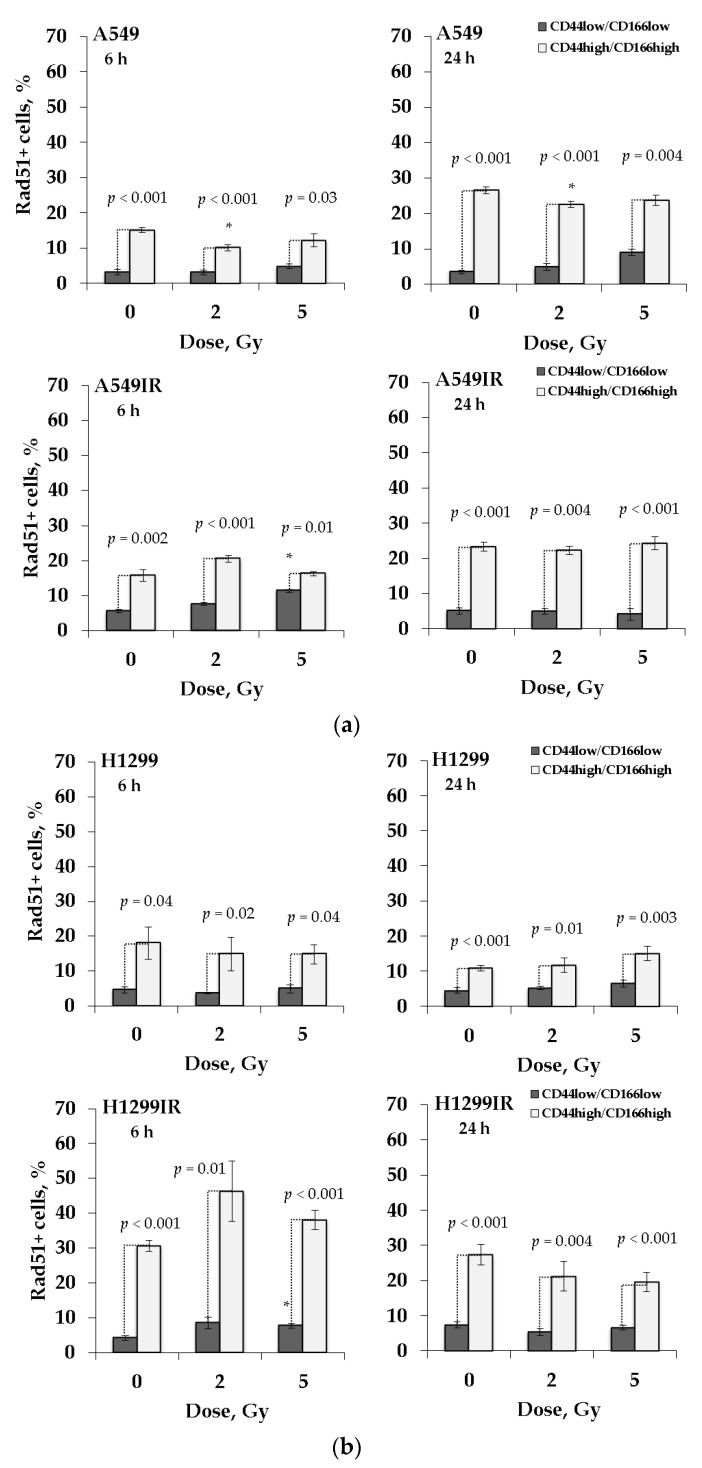
The proportion of Rad51 positive cells in the CD44low/CD166low and CD44high/CD166high populations of MFR-surviving and their parental NSCLC cells at 6 h and 24 h after exposure to 2 Gy and 5 Gy of X-ray: A549IR and A549 cells (**a**); H1299IR and H1299 cells (**b**). * *p* < 0.05. Data are means ± SEM of more than three independent experiments.

**Figure 7 ijms-22-02369-f007:**
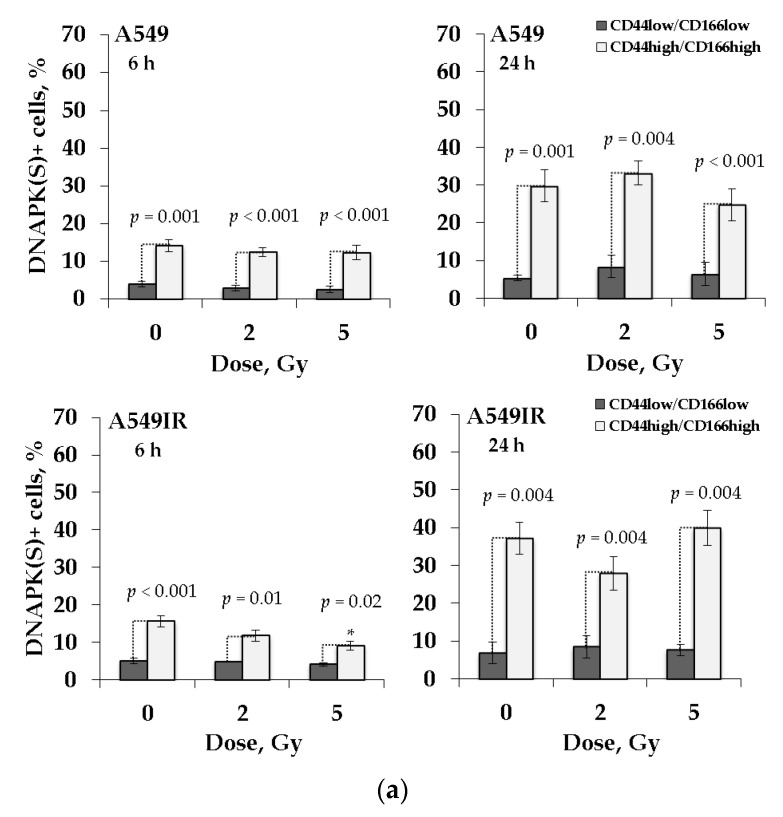
The proportion of DNA-PK(S) positive cells in the CD44low/CD166low and CD44high/CD166high populations of MFR-surviving and their parental NSCLC cells 6 h and 24 h after exposure to 2 Gy and 5 Gy of X-ray: A549IR and A549 cells (**a**); H1299 and H1299IR cells (**b**). * *p* < 0.05. Data are means ± SEM of more than three independent experiments.

**Figure 8 ijms-22-02369-f008:**
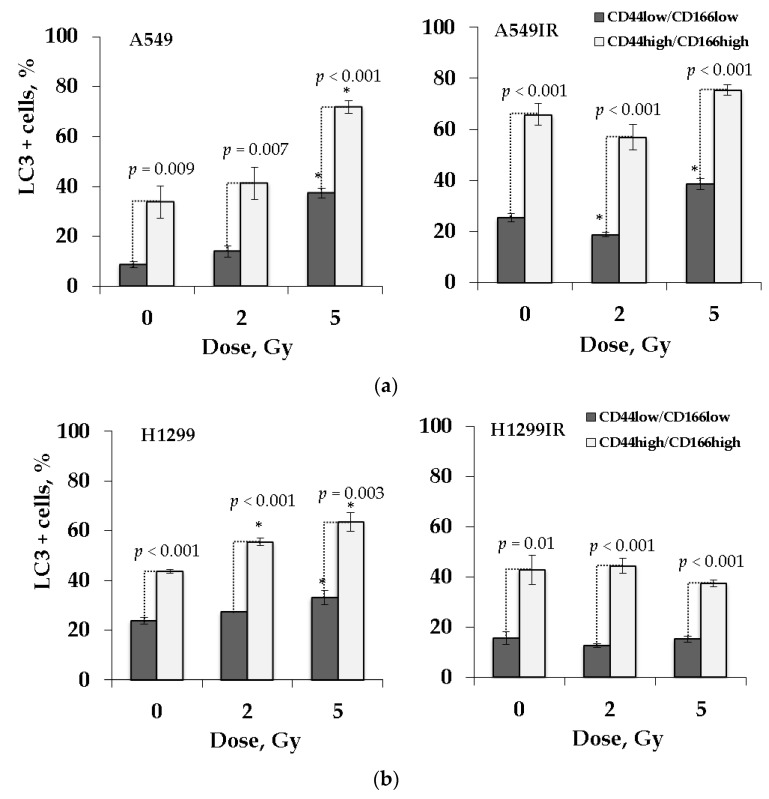
The proportion of LC3 positive cells in CD44low/CD166low and CD44high/CD166high populations of MFR-surviving and their parental NSCLC cells 24 h after exposure to 2 Gy and 5 Gy of X-ray: A549IR and A549 cells (**a**); H1299IR and H1299 cells (**b**). * *p* < 0.05. Data are means ± SEM of more than three independent experiments.

## Data Availability

Not applicable.

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
