# Peer review of "The CD44high Subpopulation of Multifraction Irradiation-Surviving NSCLC Cells Exhibits Partial EMT-Program Activation and DNA Damage Response Depending on Their p53 Status"

_ijms, 2021, doi:10.3390/ijms22052369_

Round 1

Reviewer 1 Report

NSCLC comprises 75-80% of lung malignancies and is a morbid desease. The irradiation therapy usually brings to radioresistance and it is already known that this is due to cancer stem cells.

The authors formulated the aim of their study: „..to parse out the molecular patterns of NSCLC radioresistance”,

which likely means to create a new information on the NSCLC radioresistance

Title:

CD44-high subpopulation of radioresistant NSCLC cells exhibits signs of partial EMT-program activation and DNA damage response depending on their p53 status

I do not see anything novel in the title. The title should provide a new information.

Abstract :The authors are writing at the end of it:

“Our findings suggest that radiation survived cells have a complex phenotype combining the properties of CSCs and EMT and can be considered as markers of radiotherapy response in NSCLC”.

Is it really novel? Is not it generally predictable? If really novel, this should be clearly shown in Results and Discussion.

Introduction

In introduction, the authors rely on the fresh review by Olivares-Urbano, M.A.; Grinan-Lison, C.; Marchal, J.A.; Nunez, M.I. CSC Radioresistance: A Therapeutic Challenge to Improve Radiotherapy Effectiveness in Cancer. Cells 2020, 9, doi:10.3390/cells9071651

It is about intrinsically resistant cancer stem cells and Darwinian selection of survival clones. This view is still popular but disputable.

  • There is another view that resistance is formed by transient polyploidy cells, which acquire stemness, and the cases with polyploidy comprise 36-38% in NSCLC (Gomes 2017 doi: 18632/genesandcancer.161). In the review by Olivares-Urbano there is no one single word about it, although there are the reports on A549 cells in literature highlighting the role of polyploidy in the drug-resistant response ( http://sci-hub.tw/10.1186/1475-2867-13-9;  DOI: 10.1155/2018/1754085). There is a huge literature on the role of giant cells in stemness and resistance to therapies shown for so many tumour types See, e.g.  Cancers 2018, 10, 118; doi:10.3390/cancers10040118.  Amend 2019 DOI: 10.1002/pros.23877; Moien et al 2020 https://doi.org/10.1016/j.bbcan.2020.188408 and many others, in IJMS and other journals  in last two decades.
  • The authors cannot ignore this point in their Introduction.

Methods

  • The catalogue numbers should be provided for each antibody. With some experience, I doubt that the given method of detection of RAD51 foci (with only 1h staining with the primary antibodies, in the absence of detergent during staining and washings), was sensitive enough. The authors should provide the IF pictures showing these RAD51 foci and their colocalisation with gH2AX foci.

For their aim, the authors used two lung cancer cell lines, TP-wild-type A 549 and H1299 null-type and their radioresistant sub-lines. Then they compared the reaction of these four cell lines and the reaction of to single dose 2Gy or 5Gy irradiation (0.85Gy/min), assessed after 6 and 24h for DNA repair NHEJ and HR mechanisms. They evaluated NHEJ and HR repair, also using FACS analysis for separate stem cell fractions determined as CD44/CD166-high in comparison with respectively low cell fractions.

It is not methiodically right to study the DNA repair and do not study simultaneously the cell cycle. Without that, we cannot properly compare the objects.

It is methodically wrong to study cancer cell response to irradiation in only 24 h – please see it clearly in: Illidge et al DOI: 10.1006/cbir.2000.0557; Mirzayans et al.,; Cancers 2018, 10, 118; doi:10.3390/cancers10040118.  Radioresistance, as well as resistance to genotoxic drugs,  is formed in the long time course and expresses itself (to find the difference) not after so mild dosages of irradiation and not within 24 h, the term of present studies. The behaviour during 24 h is providing some information but may mean very little, in my opinion, for prediction of further cell fate.

Results

First, the authors confirmed by WB with several marks found that isogenic radioresistant sublines were enriched by the markers of stem cells and EMT. It was also  found that HR as judged by the number of RAD51-labelled cells is higher in stem cell part of cell population . Lso this should be documented also by FACS pictures, not only bars. As well , the authors using the anti-body for both LC3I and LC3II showed at the same time points that binding of the antibody is increased (strictly saying, it can show both increase and stop of autophagy flux). It was shown that the irradiation-resistant sublines reacted to irradiation by higher autophagy, while their stem-cell-marked subpopulations did not change the response.

Discussion

 I could not find any substantially novel (for the readers of IJMS) data in the presented results and no novel information on the NSLC radioresistance was formulated in the Discussion.

My general impression

Yes, there is a difference observed and described for four settings. But it should be related to the cell cycle! We do not know from presented results, where and at which proportion are the cells in a cell cycle at 24 h in all four settings (this should make difference) and we are unaware what they are doing afterwards, at 48-96h, and on the second week. If they form giant cells, which can repair the residual DNA damage and select themselves, in which proportions and so on.

Even if we ignore this very important point, I do not see what generally new for the reader was found in these 24-h studies. If it is not so, the authors should clearly formulate the novelty in the Title, Abstract and Discussion of Results.  

My conclusion: Unfortunately, the article is outdated in the aim and the methods and subsequently it is difficult to evaluate the obtained results, their novelty remained unclear.

Reviewer 2 Report

The manuscript describes the mechanisms underlying the in vitro phenomenon of radioresistance of non-small lung cancer cells. The authors used the correct experimental procedures to analyse this phenomenon. With the regard of presentation, I would advice the authors to seek an assistance from the native English speaker, as English, in my opinion, requires further polishing.  

Minor issues: 

Abstract, line 22. Please change ‘to parse’ to ‘to establish’ or ‘to analyse’. 

Results, page 5, line 150. It does not follow from the text what the authors compared with the p-value equal to 0.04. 

Reviewer 3 Report

Pustovalova et al. studied the role of p53 for activation of NHEJ or HR after irradiation of NSCLC lines depending on their radioresistance status. They show several results but a clear “story” is missing.

Although the so-called IR lines were established of cells surviving 60 Gy a cell survival curve should be mandatory for the four cell lines. This experiment was already missing in their previous publication and thus must be presented here.

They started their Results with an analysis of gH2AX and RAD51 foci. The latter indicate the occurrence of HR. Since the amounts of RAD51 foci were very low the calculation of the amount of HR according the trapezoidal method is rather questionable.

Fig. 2 The labelling is in bad quality. What is the relevance of the analysis of the different protein expressions since they were inconsistent between both cell lines? What is the general relevance of this experiment since no further experiments about EMT were performed?

The main part of the paper is built up by experiments comparing the behaviour of CD44low/CD166low and CD44high/CD166high populations. Most of the results were independent on irradiation.

Discussion: in line 346 the authors stated that “vimentin was associated with tumor progression and metastasis H1299IR.” However, they did not show any in vivo results. So it unclear how this statement was justified.

Reviewer 4 Report

This manuscript is about the radioresistance of cells which is an important issue facing radiation-based treatments and also radiobiology. The authors investigated this point in depth. The manuscript is well written and clear to read and follow with some minor corrections required.  Such as;

in the introduction section line 94 there is a "to" missing after the word attributed

line 110 "IS" is not clear what is meant by this?

Also, there are some sections that are not clear such as;

Results section:

line 132: Does this sentence starting with "the cells were fixed..." that the cells were irradiated in suspension and then fixed?? if so the authors need to clarify and give reasons for doing that.

However, what I found difficult to accept is this:

In the discussion section, the authors stated that radiations enhance cell migration i.e. cell motilities!

Only a few references in the literature will agree with that most shows the opposite. For example, see Panzetta 2017 they explained their experimental results in detail, and also Shahhussiene et al 2019 they presented experimental data showing radiations slowing cell migration.

The question to the authors is this: did they irradiate adherent cells and found that they migrated at higher rates compared to those which are not radiated?  

I strongly think that this point needs to be resolved before considering this manuscript for publication

Round 2

Reviewer 3 Report

The authors improved their manuscipt substantially.  They even added two new figures. However, they missed to add the approprote paragraphs in the Material & Methods section. This must be made up.

The labelling in Fig. 3 (previously Fig. 2) is still baf especially in Fig. 3A.

The text should be carefully revised according to the Guide for the Use of the International System of Units (SI): https://physics.nist.gov/cuu/pdf/sp811.pdf

e. g. 2 Gy and 5 Gy instead of 2 and 5 Gy or consequently h instead of hours or 100 % instead of 100%

Reviewer 4 Report

Thanks for the comprehensive reply to that main point that I raised about the manuscript. 

However, I am sorry to say that while I am happy for the whole manuscript to be published if this particular point is solved.

Figure 4 doesn't show that there is acceleration in gap filling. And the error bars are very large, 

References which I suggested by author Shahussieni is in two issues of IJMS

I don't see how radiations would enhance cell migration?

this statement is changed I am happy to change my stand on the paper

Author Response

Reviewer comments:

Thanks for the comprehensive reply to that main point that I raised about the manuscript. 

However, I am sorry to say that while I am happy for the whole manuscript to be published if this particular point is solved.

Figure 4 doesn't show that there is acceleration in gap filling. And the error bars are very large, 

References which I suggested by author Shahussieni is in two issues of IJMS

I don't see how radiations would enhance cell migration?

this statement is changed I am happy to change my stand on the paper

Dear Reviewer, thank you for the careful reviewing of our work and valuable comments and suggestions. 

Our reply:

  • Our Fig.4 aimed to demonstrate that:
    1. p53null cells have higher migration propensity compared to p53wt cells
    2. MFR-survived p53null cells possess the delayed migration compared to their parental non-irradiated p53 null cells.
    3. MFR-survived p53wt and their parental cells had similar migratory behavior in our “scratch” assay of migration.
  • We did not make a statement that MFR increased (or accelerate) migration of neither p53 null nor p53wt NSCLC cells in our “scratch” assay of migration. We neither make a statement that radiations would enhance cell migration in our present paper.
  • Large error bars reflects data distribution between 3 sequentional cellular assays, which indicate conventional inter-assay deviation for such cell-based assays.
  • Regarding the Reviewer’s suggested reference, “author Shahussieni ..in two issues of IJMS”, we wonder whether the Reviewer misspells the author's name. We could not identify such author name in PubMed, GoogleScholar, IJMS search engines.
  • On the contrary,  the author with the name “Shahhosseini” could be found in these search engines, and his publications seemed to be relevant to the topic of our discussions if the Reviewer would mind this corrected author’s name , in particular, see:
    1. Shahhoseini, E.; Feltis, B.N.; Nakayama, M.; Piva, T.J.; Pouniotis, D.; Alghamdi, S.S.; Geso, M. Combined Effects of Gold Nanoparticles and Ionizing Radiation on Human Prostate and Lung Cancer Cell Migration. Int. J. Mol. Sci. 2019, 20, 4488. https://doi.org/10.3390/ijms20184488
    2. Shahhoseini, E.; Nakayama, M.; Piva, T.J.; Geso, M. Differential Effects of Gold Nanoparticles and Ionizing Radiation on Cell Motility between Primary Human Colonic and Melanocytic Cells and Their Cancerous Counterparts. Int. J. Mol. Sci. 2021, 22, 1418. https://doi.org/10.3390/ijms22031418

In this regard, we would emphasize that the main scope of the first paper was as stated by authors: “The chief aim of this work was to study the individual and combined effects of IR and AuNPs on cancer cell motility.” The second paper just extend these observations on another cellular models, namely, human SW48 colorectal adenocarcinoma and MM418-C1 melanoma cell lines. Even though different cell models were analyzed with respect to migration, in both papers authors rightly noted “…contradictory reports on the effects of IR on cancer cell migration [6], a few studies have shown a reduction in the rate of cell migration [11,12].” Moreover, in the second paper authors mentioned the publication on the increased motility of H1299 human non-small cell lung cancer cells [Tsutsumi, K.; Tsuda, M.; Yazawa, N.; Nakamura, H.; Ishihara, S.; Haga, H.; Yasuda, M.; Yamazaki, R.; Shirato, H.; Kawaguchi, H.; et al. Increased motility and invasiveness in tumor cells that survive 10 Gy irradiation. Cell Struct. Funct. 2009, 34, 89–96], which is in agreement with our data presented on Fig.4. In both papers, migration was related to only viability and adhesion of cancer cells up to 24 hours after acute irradiation. Our study focused on the comprehension of possible mechanisms underlying cancer cell survival after MFR treatment in relation to their p53 status.

Even our study and those two studies explored the same type of assays, we would emphasize substantial differences in experimental set-ups between our study and both papers mentioned above. First, and principle difference, in our study we performed the “scratch” test on cells that survived three weeks after sequential multi-fractionated X-ray exposure, whereas in those two papers the migration of A549 cells was analyzed for 24 hours after the acute single dose irradiation. Second, there was only 11% reduction of the migration by 24 hours of cultivation compare to the same un-irradiated cells as the controls. In our study, we compared MFR-survived cells to their parental non-irradiated cells for 24-72 hours of cultivation. Even though, we also observed the significant delay in migratory activity of p53null MFR-survived cells, but not p53wt MFR-survived cells at 24 and 48 hours.

Therefore, we would agree with the Reviewer’s suggestion and introduced the suggested references in the Discussion section as follows: “ … However, the observed spindle-shaped and rounded cell morphology of H1299IR cells was different from the typical lamellipodia morphology associated with high Vimentin [60], explaining the observed significant delay in their 1D migration activity (Fig. 4) at 24-48 h compared to their parental cells. That could be due to the substantial decrease of the adherence of these cells, which, in turn, can explicate the difference between anchorage-independent and anchorage-dependent colony formation results (Fig.1). Attractive, the acute single-dose 2 Gy and 5 Gy irradiation-induced reduction of migration and the increased adhesion of A549 cells within 24 hours after irradiation [Shahhoseini, E.; Feltis, B.N.; Nakayama, M.; Piva, T.J.; Pouniotis, D.; Alghamdi, S.S.; Geso, M. Combined Effects of Gold Nanoparticles and Ionizing Radiation on Human Prostate and Lung Cancer Cell Migration. Int. J. Mol. Sci. 2019, 20, 4488. https://doi.org/10.3390/ijms20184488]. Previously, we observed a reduction of MFR-survived A549IR cells' adhesion, albeit without any changes in their migration using the same type of migration assay in our present study (Fig.4). Such an apparent discrepancy can emphasize the significant difference in tumor cellular response to acute and multifractionated IR exposure regarding cellular migratory and adhesive properties. In this regard, our present data corroborate previous reports on increased motility of H1299 human non-small cell lung cancer cells [Tsutsumi, K.; Tsuda, M.; Yazawa, N.; Nakamura, H.; Ishihara, S.; Haga, H.; Yasuda, M.; Yamazaki, R.; Shirato, H.; Kawaguchi, H.; et al. Increased motility and invasiveness in tumor cells that survive 10 Gy irradiation. Cell Struct. Funct. 2009, 34, 89–96], supplementing other contradictory reports on the effects of IR on cancer cell migration [Moncharmont, C.; Levy, A.; Guy, J.; Falk, A.; Guilbert, M.; Trone, J.; Alphonse, G.; Gilormini, M.; Ardail, D.; Toillon, R.; et al. Radiation-Enhanced Cell Migration/Invasion Process: A Review. J. Crit. Rev. Oncol. 2014, 92, 133–142; Akino, Y.; Teshima, T.; Kihara, A.; Kodera-Suzumoto, Y.; Inaoka, M.; Higashiyama, S.; Furusawa, Y.; Matsuura, N. Carbon-Ion Beam Irradiation Effectively Suppresses Migration and Invasion of Human Non-Small-Cell Lung Cancer Cells. Int. J. Radiat. Oncol. Biol. Phys. 2009, 75, 475–481; Ogata, T.; Teshima, T.; Inaoka, M.; Minami, K.; Tsuchiya, T.; Isono, M.; Furusawa, Y.; Matsura, N. Carbon Ion Irradiation Suppresses Metastatic Potential of Human Non-small Cell Lung Cancer Cells through the Phosphatidylinositol-3-Kinase/ Akt Signaling Pathway. J. Rdiat. Res. 2011, 52, 374–379.].  In sum, p53 status significantly affected the 1D confined migratory behavior (wound healing) and an irradiated cell's capability to continue to divide and form a colony of NSCLC cells before and after MFR.”

Round 3

Reviewer 4 Report

I am happy now for the manuscript to be published as that point now is cleared